# Single-Cell RNA Sequencing Analysis for Oncogenic Mechanisms Underlying Oral Squamous Cell Carcinoma Carcinogenesis with *Candida albicans* Infection

**DOI:** 10.3390/ijms23094833

**Published:** 2022-04-27

**Authors:** Yi-Ping Hsieh, Yu-Hsueh Wu, Siao-Muk Cheng, Fang-Kuei Lin, Daw-Yang Hwang, Shih-Sheng Jiang, Ken-Chung Chen, Meng-Yen Chen, Wei-Fan Chiang, Ko-Jiunn Liu, Nam Cong-Nhat Huynh, Wen-Tsung Huang, Tze-Ta Huang

**Affiliations:** 1Institute of Basic Medical Sciences, College of Medicine, National Cheng Kung University, Tainan 701401, Taiwan; pin1000822@gmail.com; 2Institute of Oral Medicine, College of Medicine, National Cheng Kung University, Tainan 701401, Taiwan; yuhsueh1107@hotmail.com (Y.-H.W.); meitung0703@gmail.com (F.-K.L.); omsboy@gmail.com (K.-C.C.); ccdc0002.tw@gmail.com (M.-Y.C.); 3Department of Stomatology, National Cheng Kung University Hospital, College of Medicine, National Cheng Kung University, Tainan 701401, Taiwan; 4National Institute of Cancer Research, National Health Research Institutes, Tainan 70456, Taiwan; isaaccheng86@gmail.com (S.-M.C.); dawyanghwang@nhri.edu.tw (D.-Y.H.); kojiunn@nhri.edu.tw (K.-J.L.); 5National Institute of Cancer Research, National Health Research Institutes, Zhunan 35053, Taiwan; ssjiang@nhri.edu.tw; 6Chi Mei Medical Center, Tainan 71004, Taiwan; bigfanfan@yahoo.com.tw; 7School of Dentistry, National Yang Ming University, Taipei 11221, Taiwan; 8Graduate Institute of Medicine, College of Medicine, Kaohsiung Medical University, Kaohsiung 807377, Taiwan; 9School of Medical Laboratory Science and Biotechnology, Taipei Medical University, Taipei 110301, Taiwan; 10Institute of Clinical Pharmacy and Pharmaceutical Sciences, Institute of Clinical Medicine, National Cheng Kung University, Tainan 704302, Taiwan; 11Laboratory of Oral-Maxillofacial Biology, Faculty of Odonto-Stomatology, University of Medicine and Pharmacy at Ho Chi Minh City, Ho Chi Minh City 749000, Vietnam; namhuynh@ump.edu.vn

**Keywords:** tumor heterogeneity, tumor microenvironment, oral squamous cell carcinoma, oral potentially malignant disorder, *Candida albicans*, single-cell RNA sequencing analysis

## Abstract

Oral squamous cell carcinoma (OSCC) carcinogenesis involves heterogeneous tumor cells, and the tumor microenvironment (TME) is highly complex with many different cell types. Cancer cell–TME interactions are crucial in OSCC progression. *Candida albicans* (*C. albicans*)—frequently pre-sent in the oral potentially malignant disorder (OPMD) lesions and OSCC tissues—promotes malignant transformation. The aim of the study is to verify the mechanisms underlying OSCC car-cinogenesis with *C. albicans* infection and identify the biomarker for the early detection of OSCC and as the treatment target. The single-cell RNA sequencing analysis (scRNA-seq) was performed to explore the cell subtypes in normal oral mucosa, OPMD, and OSCC tissues. The cell composi-tion changes and oncogenic mechanisms underlying OSCC carcinogenesis with *C. albicans* infec-tion were investigated. Gene Set Variation Analysis (GSVA) was used to survey the mechanisms underlying OSCC carcinogenesis with and without *C. albicans* infection. The results revealed spe-cific cell clusters contributing to OSCC carcinogenesis with and without *C. albicans* infection. The major mechanisms involved in OSCC carcinogenesis without *C. albicans* infection are the IL2/STAT5, TNFα/NFκB, and TGFβ signaling pathways, whereas those involved in OSCC carcinogenesis with *C. albicans* infection are the KRAS signaling pathway and E2F target down-stream genes. Finally, stratifin (SFN) was validated to be a specific biomarker of OSCC with *C. albicans* infection. Thus, the detailed mechanism underlying OSCC carcinogenesis with *C. albicans* infection was determined and identified the treatment biomarker with potential precision medicine applications.

## 1. Introduction

Cancer evolves dynamically and continuously. During carcinogenesis, tumors become increasingly heterogeneous. A single tumor mass has several cell subgroups, including cells with distinct cellular morphology, gene expression, proliferation, and metastatic potential [1]. Genomic instability and epigenetic deregulation contribute to tumor heterogeneity [2], a characteristic noted in most patients with cancer. High levels of tumor heterogeneity can lead to a poor prognosis [3]. Tumor heterogeneity can considerably impede cancer therapy because it can result in a tumor having cancer cells with varying levels of sensitivity to anticancer interventions. Therapeutic intervention for tumors with heterogeneity can increase the cancer cell subpopulations that are resistant to therapy. This also is a common cause of recurrence in previously responsive patients with cancer and thus leads to poor outcomes [4].

The tumor microenvironment (TME) is also a major factor affecting increases in cancer cell heterogeneity and contributing to cancer progression [5,6]. TME consists of heterogeneous cell populations, including immune cells, endothelial cells, fibroblasts, signaling molecules, and the extracellular matrix. A tumor is closely associated with its surrounding microenvironment and communicates with it constantly [7]. Immune cells are critical TME components. Immune cell infiltration can affect tumor growth, progression, therapeutic outcomes, and patient prognosis [8,9]. Tumor-derived endothelial cells (TECs) differentiate from cancer cells and build up the inner layer of the blood vessels in growing tumors [10]. Compared with normal endothelial cells, TECs are more similar to tumor cells with disturbed morphology and phenotypes. The TECs promote tumor angiogenesis, metastasis, and drug resistance [11,12,13]. Cancer-associated fibroblasts also contribute to carcinogenesis by increasing cancer cell proliferation, migration, and viability [14,15]. These distinct cell subpopulations interact closely with tumor cells and are involved in tumor progression. The cell subgroups that change in the tumor and the neighboring microenvironment during cancer progression may be new targets for early cancer detection and precision cancer therapy.

Oral squamous cell carcinoma (OSCC) is a common cancer type worldwide [16]. Cigarette smoking, alcohol consumption, and betel quid chewing are its major risk factors; other factors that can lead to OSCC include excessive sun exposure, viral or fungal infection, poor nutrition, and poor oral hygiene [17]. Patients with OSCC are treated using surgery, chemotherapy, radiotherapy, immune therapy, or a combination of these. The treatment modality used depends on the clinical stage of OSCC, with neck lymph node metastasis being a key consideration. Patients often present lymph node metastasis at the late stage of OSCC. However, the average five-year survival rate of patients with advanced OSCC is approximately 40% [18]. The carcinogenesis of OSCC occurs gradually and oral epithelial dysplasia is considered a potential histologic precursor of OSCC. Most OSCC cases are preceded by clinically evident oral potentially malignant disorders (OPMDs) [19]. OPMDs include oral leukoplakia, erythroplakia, erythroleukoplakia, and oral submucous fibrosis, all with a high potential for malignant transformation [20]. *Candida albicans* is a common pathogenic fungus in the oral cavity. Individuals with poor oral hygiene, immunodeficiency, or long-term antibiotic use are prone to developing *C. albicans* infections. The pathological pattern of chronic hyperplastic candidiasis is accompanied by pseudoleukoplakia symptoms [21]. Chronic candidiasis, classified as an OPMD, demonstrates a high rate of malignant transformation [22]. The prevalence of oral *C. albicans* infection is high in patients with OSCC and OPMD [23]. *C. albicans* has been linked to the OSCC and OPMD etiopathogeneses [24]. Patients with *C. albicans*-infected lesions present progressively more severe dysplasia than do patients without. Leukoplakia cases with *C. albicans* infection lead to a higher rate of malignant transformation to OSCC than do uninfected leukoplakia cases [25]. *C. albicans* can produce carcinogenic compounds, such as N-nitrosobenzyl methylamine and acetaldehyde, which bind to DNA and form adducts that cause genomic instability—resulting in oncogene formation and cancer development initiation [26]. *C. albicans* infection leads to proinflammatory cytokine upregulation and immune cell recruitment, promoting OSCC carcinogenesis [27,28]. However, the mechanisms underlying the *C. albicans* infection-led promotion of malignantly transformation to OSCC and OPMD remain under investigation.

Cancer studies thus far have used numerous cancer cell types to explore gene expression and regulation in general [29,30]. However, tumors are heterogeneous, composed of malignant cells, immune cells, and stromal cells. When studying several cancer cells simultaneously, the gene expression of an important subtype may be easily overlooked; moreover, investigating the interaction between different cell subtypes can be difficult. Single-cell analysis can be a powerful approach to distinguishing different cell subgroups in a tumor mass. Therefore, such analysis can be used to identify rare cell subpopulations, explore TME characteristics and cancer cell interactions, and determine the cell subpopulations change during cancer progression [31]. In the current study, single-cell RNA sequencing (scRNA-seq) was used to investigate OSCC’s malignant transformation with and without *C. albicans* infection. Our results highlight the cell subpopulations that change during OSCC progression as well as the related cancer cell–TME interactions.

## 2. Results

### 2.1. Single-Cell RNA Sequencing in the Normal Tissue, OPMD Lesion with C. albicans Infection, OSCC Tissue with C. albicans Infection, and OSCC Tissue without C. albicans Infection

Here, we included one clinical tissue specimen from normal oral mucosa, an OPMD lesion with *C. albicans* infection, OSCC tumor without *C. albicans* infection, and OSCC tumor with *C. albicans* infection. We isolated a total of 18,433 cells (Appendix A), on which we conducted clustering and cell typing analyses, in addition to identifying differentially expressed features (Appendix A). According to the t-distributed stochastic neighbor embedding (tSNE) results, these cells could be divided into immune cells and cancer-related cells according to their typical biomarkers (Figure 1a). Here, the biomarkers were identified using single-cell transcriptomic analysis of head and neck cancer [32].

The cell subtypes could be divided into 23 clusters (Figure 1b); of these, clusters 2, 7, 12, 13, 16, and 19 included cancer-related cells, whereas the remaining 17 clusters contained immune cells. These cells are in the four main groups—namely normal tissue, OPMD lesion with *C. albicans* infection, OSCC tissue with *C. albicans* infection, and OSCC tissue without *C. albicans* infection—were visualized through tSNE (Figure 1c) and classified on the basis of their specific biomarkers in each cluster (Figure 1d).

### 2.2. Cell Composition Change in the Normal Tissue, OPMD Lesion with C. albicans Infection, OSCC Tissue with C. albicans Infection, and OSCC Tissue without C. albicans Infection

To investigate cell composition changes during carcinogenesis, we subclustered the cells. The cancer-related cells were divided further into 13 clusters (Figure 2a) and then classified into various cell types according to their typical biomarkers: epithelial cells (clusters 4, 9, and 12), endothelial cells (clusters 6, 8, and 11), fibroblasts (clusters 0, 1, 2, 3, 5, and 7), and other cell types (cluster 10; Figure 2b). Subsequently, immune cells were divided further into 15 clusters (Figure 3a) and then classified into various cell types according to their typical biomarkers: B cells (clusters 3 and 6), macrophages (clusters 5, 8, and 10), mast cells (cluster 12), neutrophils (cluster 11), T cells (clusters 0, 1, 2, 4, and 9), and unknown cell type (clusters 7, 13, and 14; Figure 3b). The cancer-related and immune cells in the four main groups were visualized through tSNE (Figure 2c and Figure 3c, respectively) and classified on the basis of their specific biomarkers in each cluster d and Figure 3d, respectively; Appendix A).

The cell ratios of the clusters in the four main groups were then analyzed. The number of cluster 0, 1, and 3 fibroblasts was significantly lower in the OPMD lesion and OSCC tissues than in the normal tissue. Nevertheless, compared with that in the normal tissue, the number of cluster 2 fibroblasts was significantly higher in the OSCC tissue without *C. albicans* infection and that of cluster 5 and 7 fibroblasts was higher in the OPMD lesion with *C. albicans* infection. Moreover, compared with that in the normal tissue, the number of cluster 4, 9, and 12 epithelial cells was significantly higher in the OSCC tissue with *C. albicans* infection and the numbers of cluster 8 endothelial cells were significantly higher in the OPMD lesion and OSCC tissues. These results indicated that cluster 5 and 7 fibroblasts were related to the malignant transformation in OPMD lesions with *C. albicans* infection, that cluster 2 fibroblasts may play an important role in OSCC carcinogenesis without *C. albicans* infection, and cluster 4, 9, and 12 epithelial cells are major cell types in OSCC tissues with *C. albicans* infection (Appendix A and Figure 4a).

The number of cluster 0 T cells was significantly lower in the OSCC tissue (partic-ularly that with *C. albicans* infection) than in the normal tissue. However, the number of cluster 0, 1, and 2 T cells was significantly lower in the OSCC tissue with *C. albicans* infection than on the OPMD lesion with *C. albicans* infection. Moreover, the number of cluster 4 T cells was significantly lower in the OSCC tissue without *C. albicans* infection than in the normal tissue. In addition, the number of cluster 3 and 4 B cells were lower in the OSCC tissues than in the normal tissue, whereas the number of cluster 5 and 10 macrophages was significantly higher in OSCC tissues with *C. albicans* infection than in the normal tissue. The number of cluster 11 neutrophils was also higher in the OSCC tissue with *C. albicans* infection. These results revealed that cluster 4 T cells may correlate with OSCC carcinogenesis without *C. albicans* infection. Moreover, the reduction in the number of cluster 0, 1, and 2 T cells may be a factor contributing to the promotion of malignant transformation due to *C. albicans* infection. Furthermore, cluster 5 and 10 macrophages may play a crucial role in OSCC carcinogenesis with *C. albicans* infection. (Appendix A and Figure 4b).

### 2.3. Regulatory Pathways Involved in OSCC Carcinogenesis with and without C. albicans Infection

To analyze the mechanisms underlying OSCC carcinogenesis with or without *C. albicans* infection, the Molecular Signatures Databases were used to perform Gene Set Variation Analysis (GSVA) in the four tissue groups. The genes significantly upregulated in the cancer-related cells of the OSCC tissue with *C. albicans* infection were KRAS_SIGNALING_DN and E2F_TARGETS (Figure 5). On the contrary, in the cancer-related cells of the OSCC tissue without *C. albicans* infection, IL2_STAT5_SIGNALING, TNFA_SIGNALING_VIA_NFKB, and TGF_BETA_SIGNALING were significantly upregulated; these genes are involved in myogenesis, ultraviolet (UV) response, epithelial–mesenchymal transition, and coagulation (Figure 5a).

Among the immune cells, REACTIVE_OXYGEN_SPECIES_PATHWAY and KRAS_SIGNALING_UP were significantly upregulated in the OSCC tissue with *C. albicans* infection; these genes contribute to coagulation, epithelial–mesenchymal transition, angiogenesis, and cholesterol homeostasis. On the contrary, in the OSCC tissue with *C. albicans* infection, KRAS_SIGNALING_DN, WNT_BETA_CATNIN_SIGNALING, and TGF_BETA_SIGNALING were upregulated; these genes are involved in allograft rejection, interferon-gamma response, UV response, and G2M checkpoint (Figure 5b).

Taken together, these results revealed that the mechanisms underlying OSCC carcinogenesis due to *C. albicans* infection are different from those underlying OSCC carcinogenesis without *C. albicans* infection. The IL2/STAT5, TNFα/NFκB, and TGFβ signaling pathways may be essential for promoting OSCC carcinogenesis without *C. albicans* infection; moreover, immune cells with high expression of KRAS, WNT/β-catenin, and TGFβ signaling pathway genes may contribute to this promotion. On the contrary, in OSCC carcinogenesis with *C. albicans* infection, the KRAS signaling pathway and E2F target downstream genes may be the major mechanisms involved; here, the upregulation of ROS and KRAS signaling pathway genes in immune cells correlated to the promotion of OSCC progression.

### 2.4. Stratifin (SFN) as the Specific Biomarker of OSCC with C. albicans Infection

Our scRNA-seq analysis results indicated that SFN may be the specific biomarker for cluster 4 and 9 cells, the number of which was significantly higher in the OSCC tissue with *C. albicans* infection. Therefore, we analyzed whether SFN has the potential to be a specific biomarker of OSCC with *C. albicans* infection. The SFN expression in the normal tissue, OPMD lesion, OSCC tissue with *C. albicans* infection, and OSCC tissue without *C. albicans* infection were analyzed. The IHC staining results revealed that SFN expression was significantly higher in the tissues with *C. albicans* infection, especially in the OSCC tis-sue—suggesting that SFN can specifically predict malignant transformation due to *C. albicans* infection (Figure 6).

## 3. Discussion

Cancer studies have relied on measuring the average of bulk tumors to identify oncogenic mutations, aberrant regulatory programs, and disease subtypes in the tumors. However, tumors are intricate in that they comprise a diverse set of cells, including malignant cells, immune cells, and stromal cells. Their interactions within the TME are critical to various aspects of carcinogenesis. The diversity among the malignant, stromal, and immune cells leads to immune–cancer cell and stromal–cancer cell interactions, which are affected by spatial dynamics and clonal evolution. Therefore, when studying several cancer cells simultaneously, the essential roles of the specific subtypes may be overlooked, hampering the identification of the evolution of the specific subtypes during carcinogenesis [33,34]. Notably, the composition changes in the OSCC tumors and TME can affect cancer development, progression, and metastasis. Nevertheless, elucidating the detailed mechanism underlying OSCC tumor composition and malignant cell–TME interaction may revolutionize OSCC diagnosis and treatment.

Single-cell expression profiling studies provide a new approach for distinguishing different cell subgroups in tumor masses. The classic approaches used for single-cell isolation are micromanipulation, fluorescent-activated cell sorting, and laser capture microdissection. However, these methods require advanced instrumentation but result in low productivity and tend to exhibit cell contamination. Microfluidic technology, a highly efficient approach, allows a group of cells to be sequentially passed through an aperture. Each drop that passes through the aperture contains only one cell with the reverse transcription reagents and an identifier [35]. Conventional bulk RNA expression analysis provides an average of profiles in the cell mixture, thus helping to avoid the consequent loss of critical information on cellular heterogeneity. scRNA-seq—a novel technology that enables deep cellular characterization of the RNA profiles one cell at a time—has been widely applied in various research fields. In cancer research, scRNA-seq has been applied to identify cell composition in tumor tissues, understand how malignant cells transform from normal cells, survey cell communication within cancer cells and the TME, and investigate the cellular response to genetic manipulation or drugs [36]. In lung cancer research, scRNA-seq has revealed the malignant cell development process as well as how the lymphocytes infiltrate after immune therapy [37]. In breast cancer research, it has been applied to cluster cell populations in breast tumors with different molecular subtypes to identify distinct populations possibly related to poor prognosis and drug resistance [38,39]. In the current study, we used a 10× Chromium system and NGS to explore the cell composition changes within OSCC carcinogenesis as well as the mechanisms underlying OSCC carcinogenesis with or without *C. albicans* infection.

*C. albicans* is a common yeast-like opportunistic pathogen in the oral cavity. According to the host defense mechanisms or TME in the oral cavity, *C. albicans* can transform into a pathogenic organism, leading to oral mucosal infection. *C. albicans*, the primary cause of oral candidiasis, has been identified to be correlated with OSCC development. The association between oral *C. albicans* infection and OSCC or OPMD is noteworthy. OPMD patients with *C. albicans* infection have a higher rate of malignant transformation [40]. *C. albicans* produce hydrolysis-related enzymes and virulence factors to invade tissues. *C. albicans* invasion, then promotes a hyperplastic epithelial response in the oral mucosal cells, making them dysplastic and transforming into carcinoma [41]. *C. albicans* also produces carcinogenic compounds, such as nitrosamines and N-nitroso benzyl methylamine, which cause genomic instability. As such, *C. albicans* infection enhances OSCC progression by increasing oncometabolite concentration, matrix metalloproteinases production, oncogenic signaling pathways, and metastasis-associated genes. In addition, in epithelial cells, *C. albicans* infection activates the PI3K/Akt/NFκB, p38/JNK, and ERK1/2/MAPK signaling pathways, which result in the secretion of an array of antimicrobial peptides, alarmins, and proinflammatory cytokines. The released proinflammatory cytokines attract immune cells, mainly neutrophils, macrophages, and T helper 17 cells, to the infection foci [42,43]. In animal studies, *C. albicans* has been identified to promote OSCC progression through inflammation, induce metastatic gene overexpression, and cause considerable changes in epithelial–mesenchymal transition markers. Cancer patients with *C. albicans* infection can also demonstrate immune status alteration [44]. In the current study, our scRNA-seq analysis revealed that immune cell composition in the OSCC tissue with *C. albicans* infection differed from that of the OSCC tissue without *C. albicans* infection: The number of cluster 0 T cells was significantly lower in OSCC tissues (particularly the tissue with *C. albicans* infection) than in the normal tissue. Moreover, the number of cluster zero, one, and two T cells were significantly lower in the OSCC tissue with *C. albicans* infection than in the OPMD lesion with *C. albicans* infection, and the number of cluster four T cells was significantly lower in the OSCC tissue without *C. albicans* infection than in the normal tissue. On the contrary, the number of cluster 5 and 10 macrophages was significantly higher in OSCC tissues with *C. albicans* infection than in the normal tissue. The number of cluster 11 neutrophils was also higher in the OSCC tissue with *C. albicans* infection. Taken together, our results revealed that immune cell infiltration is involved in carcinogenesis due to *C. albicans* infection.

The IL2/STAT5 signaling pathway regulates many immune functions, particularly those involved in regulatory T cell development or function [45]. In various cancers, aberrant STAT5 signaling increases the expression of target genes, such as cyclin D, Bcl-2, and MMP-2, eventually resulting in the promotion of cell proliferation, survival, and metastasis [46]. The TNFα/NFκB signaling pathway might play a critical role in the development of the proinflammatory condition in many cancers. The expression of this pathway is upregulated in macrophages because they generate several inflammatory cytokines such as TNFα, IL-1, and IL-6. These cytokines recruit more inflammatory cells to the TME to enhance tumor cell proliferation and survival. The TNFα/NFκB pathway is also strongly expressed in epithelial cells to directly facilitate their proliferation and survival [47]. TGFβ is critical in the regulation of embryogenic development, inflammation, tissue repair, and the maintenance of adult tissue homeostasis. TGFβ signaling pathway dysregulation has been identified in many cancers, and TGFβ signaling is a key factor that promotes epithelial–mesenchymal transition and cancer resistance to chemotherapy [48]. In the current study, we found that the IL2/STAT5, TNFα/NFκB, and TGFβ signaling pathways are essential mechanisms promoting OSCC carcinogenesis without *C. albicans* infection. However, the mechanisms underlying OSCC carcinogenesis with *C. albicans* infection differed from those underlying OSCC carcinogenesis without *C. albicans* infection. In OSCC carcinogenesis with *C. albicans* infection, the major mechanisms involved are the KRAS signaling pathway and E2F target downstream genes.

SFN is a member of the 14-3-3 family of highly conserved soluble acidic proteins that regulate various biological functions—which include cell cycling, growth, development, survival, and death as well as gene transcription. SFN, strongly expressed in many cancers, is involved in carcinogenesis through the regulation of cell proliferation, differentiation, and death in tumors [49,50]. In the current study, SFN was found to be strongly expressed in the tissue with *C. albicans* infection, particularly the OSCC tissue. Studies have demonstrated that SFN is downstream of the E2F target gene and that SFN expression is relatively high in hepatocellular carcinoma cells with E2F overexpression and TP53 mutation [51,52]. SFN may thus serve as the specific biomarker for predicting malignant transformation due to *C. albicans* infection. In the future, we will further explore the mechanism of SFN involves in the oral squamous cell carcinoma carcinogenesis with *C. albicans* infection. SFN has the potential to develop as an OSCC biomarker, which could be developed for precision medicine applications.

In this research, there is only one specimen collected in each group of normal oral mucosa, OPMD with *Candida albicans* infection, OSCC with *C. albicans* infection, and OSCC without *C. albicans* infection. The individual genetic background could not be eliminated due to only one patient recruited in each group. However, due to omics data generated by scRNA-seq, a total of 18,433 cells were isolated for clustering, cell typing analyses, and identifying differentially expressed features. The major mechanisms involved in OSCC carcinogenesis with or without *C. albicans* infection could be identified. The other limitation of this research is that our result could not be solidated due to *C. albicans* infection oral mucosa and OPMD lesion without *C. albicans* infection were not recruited. It is because there is no need for surgical intervention for the patient’s oral mucosa with *C. albicans* infection and difficulty finding the OPMD lesion without *C. albicans* infection due to the prevalence of oral *C. albicans* infection is high in patients with OPMD [23]. However, the major mechanisms involved in OSCC carcinogenesis without *C. albicans* infection are the IL2/STAT5, TNFα/NFκB, and TGFβ signaling pathways, whereas those involved in OSCC carcinogenesis with *C. albicans* infection are the KRAS signaling pathway and E2F target downstream genes were revealed in this research. The stratifin (SFN) was also validated to be a specific biomarker of OSCC with *C. albicans* infection. The detailed mechanism underlying OSCC carcinogenesis with *C. albicans* infection was determined in this research and identified the treatment biomarker with potential precision medicine applications.

## 4. Materials and Methods

### 4.1. Clinical Patient Specimen Collection

This study was approved by the Institutional Review Board of National Cheng Kung University Hospital (IRB No.: B-ER-108-424) and complied with medical research protocols outlined in the Declaration of Helsinki by the World Medical Association.

The included patients underwent scheduled excision surgery. We collected one specimen each of the OPMD lesion with *C. albicans* infection, and OSCC tissue with or without *C. albicans* infection. Normal tissue is the healthy gingival mucosa that was collected when the healthy patient received wisdom teeth extraction. Thereafter, tissue dissociation and further analysis were performed.

### 4.2. Candidiasis Detection

To investigate cell composition change in OSCC tissue and the oncogenic mechanism of *C. albicans* infection during OSCC carcinogenesis, the normal tissue, OPMD lesion with *C. albicans* infection, OSCC tissue with *C. albicans* infection, and OSCC tissue without *C. albicans* infection was used in scRNA-seq analysis (Appendix A).

Before clinical specimen collection, the patients were tested for *C. albicans* infection (Appendix A) by using CHROMagar *Candida* (Becton Dickinson, Franklin, NJ, USA)—which is a medium containing chromogenic substrates that react with enzymes produced by different pathogens and produce colonies of varying color and morphology, with the appearance of green colonies indicating *C. albicans* positivity [53]. In brief, the samples for *C. albicans* testing were collected using a dry sponge swab to softly rub the surface of normal buccal mucosa, OPMD lesion, or tumor tissue. The samples were cultured on CHROMagar Candida and incubated at 35 °C for 2–4 days. We then observed the color production and morphology of the colonies to determine whether the patients had *C. albicans* infection (i.e., candidiasis).

### 4.3. Tissue Dissociation

Following candidiasis detection and patient treatment, the normal tissue, OPMD lesion, and OSCC tissues were collected. The specimens were rinsed with 1× phosphate-buffered saline (PBS; VWR, Philadelphia, PA, USA). Tissue dissociation was performed using the human tissue dissociation kit from Miltenyi Biotec (Bergisch Gladbach, Germany) according to the manufacturer’s protocol. Tissues were sliced into small pieces with a scalpel and soaked in gentleMACSTM C Tubes (Miltenyi Biotec, Bergisch Gladbach, Germany), containing Enzyme H, Enzyme R, and Enzyme A (all provided in the human tumor dissociation kit) as well as a cell culture medium (i.e., Dulbecco-modified Eagle’s medium; Gibco, New York, NY, USA), neutral protease Dispase II (1.08 U/mg; Sigma-Aldrich, St. Louis, MO, USA), and collagenase IV (0.1 mg/mL; Sigma-Aldrich, St. Louis, MO, USA). The C tube was processed on an automatic tissue homogenization/single-cell separator (gentleMACSTM Octo Dissociator with Heaters; Miltenyi Biotec, Bergisch Gladbach, Germany) and maintained at 37 °C for 10 min to separate tissues into single cells. Thereafter, the C tube was left to stand at room temperature for 10 min. The resulting suspension was passed through a 40-micrometer cell strainer. The cell-containing suspension was collected, and 1× red blood cell lysis buffer (BD) was added. After red blood cell lysis was completed, the suspension was centrifuged at 1500 rpm for 5 min; subsequently, the supernatant was removed, and the spun-down cells were washed with PBS containing 0.04% bovine serum albumin (Cyrusbioscience, Taipei, Taiwan) and centrifuged again. Next, the enriched cells were counted with trypan blue (VWR, Philadelphia, PA, USA), and an automatic cell counter was used to determine the number of viable cells. Finally, the concentration was adjusted to 1000 cells/μL for microfluidic cell sorting.

### 4.4. 3′ Gene Expression Library Construction on 10× Genomics Platform

We performed 3′ gene expression library construction on Chromium Next GEM Single Cell 3′ Reagent Kits v3.1 (10× Genomics, Pleasanton, CA, USA). First, we mixed an appropriate volume of nuclease-free water, reagent, partitioning oil, beads, and enzyme to Chromium Next GEM Chip G with a corresponding volume of single-cell suspension to generate a single-cell gel beads-in-emulsion (GEM). Subsequently, to break the GEMs, generate barcoded, and synthesize full-length complementary DNA (cDNA), we used the following polymerase chain reaction (PCR) program: 53 °C for 45 min, 85 °C for 5 min, and end at 4 °C. Thereafter, using DynaBeads MyOne silane magnetic beads, we purified first-strand cDNA. Each cDNA was then amplified using the following PCR program: 98 °C for 3 min, followed by 10 cycles of 98 °C for 15 s, 67 °C for 20 s, and 72 °C for 1 min; 1 cycle at 72 °C for 1 min; and end at 4 °C. Thereafter, we used the SPRIselect Reagent Kit (Beckman Coulter, Brea, CA, USA) to clean up. Third, a sample index i7, P5, P7, and read 2 primer sequences were added through enzymatic fragmentation, end repair, A-tailing, adaptor ligation, and PCR. Qubit dsDNA HS (Invitrogen, Waltham, MA, USA) was then used to quantify the library construction [54].

### 4.5. Next-Generation Sequencing

NextSeq 550 Sequencing System and NextSeq 500/550 High-Output v2.5 Kit (Illumina, San Diego, CA, USA) were used to perform next-generation sequencing (NGS) of the 3′ gene expression library. The parameters for paired-end, single indexing sequencing were as follows: Read 1, 28 cycles; i7 Index, 8 cycles; Read 2, 91 cycles; and Library Loading NextSeq 500/550, 1.5 pM [55].

### 4.6. Bioinformatic Analysis of NGS Data

After NGS, sequenced reads from all libraries were aligned and quantified using Cell Ranger Single Cell Software Suite (version 3.1.0; 10× Genomics, Pleasanton, CA, USA) and STAR (RRID: SCR_005622) against the 10× Genomics prebuilt GRCh38 reference genome [56]. Quality control analyses were then performed to eliminate the low-quality cells. Cells that contain very few reads, low genome mapping ratios, high mitochondrial mapping ratios, >10% mitochondrial genes, or <200 nFeature_RNA were eliminated.

### 4.7. Immunohistochemistry Staining

We obtained paraffin-embedded specimens of normal tissue, OPMD lesion with *C. albicans* infection, and OSCC tissue with and without *C. albicans* infection from the Department of Pathology at National Cheng Kung University Hospital. These specimens were sectioned and then subjected to immunohistochemistry (IHC) analysis with the Mouse/Rabbit Probe HRP Labeling Kit with DAB Brown Kit (BioTnA, Kaohsiung, Taiwan), performed according to the manufacturer’s instructions. Moreover, a stratifin (SFN) antibody (Abcam, Cambridge, UK) was used to analyze SFN expression in the tissue sections. To control for the staining variability of different batches of experiments, we included a negative control in each experiment.

## 5. Conclusions

The specific cell clusters contributing to OSCC carcinogenesis with and without *C. albicans* infection were revealed in this research. Cluster five and seven fibroblasts are involved in OPMD lesions with *C. albicans* infection, cluster 2 fibroblasts promote OSCC carcinogenesis without *C. albicans* infection, and cluster 4, 9, and 12 epithelial cells are major cell types in OSCC tissues with *C. albicans* infection. Furthermore, cluster 5 and 10 macrophages were found to play a crucial role in OSCC carcinogenesis with *C. albicans* infection. The GSVA data demonstrated that the IL2/STAT5, TNFα/NFκB, and TGFβ signaling pathways are the essential mechanisms promoting OSCC carcinogenesis without *C. albicans* infection and the KRAS signaling pathway and E2F target downstream genes are the major mechanisms promoting OSCC carcinogenesis with *C. albicans* infection. The SFN was also validated to be a specific biomarker of OSCC with *C. albicans* infection.

## Figures and Tables

**Figure 1 ijms-23-04833-f001:**
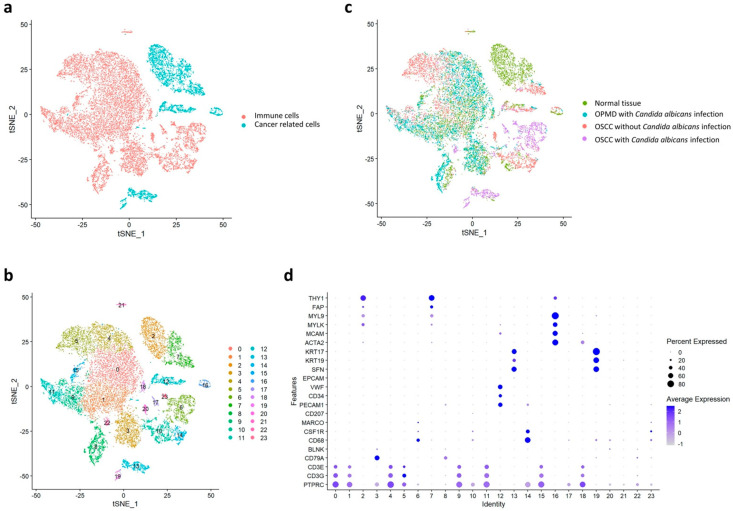
Visual distribution of dimensionality reduction in all cell populations. (**a**) Cell distribution of all samples generated through *t*-distributed stochastic neighbor embedding. The cells are divided into cancer-related cells and immune cells. (**b**) Classification of cells into 24 subtypes by using the Louvain algorithm. (**c**) Separation of the four major groups: normal tissue, OPMD lesion with *Candida albicans* infection, OSCC tissue with *C. albicans* infection, and OSCC tissue without *C. albicans* infection. (**d**) Classification of cell types in each cluster by using specific biomarkers.

**Figure 2 ijms-23-04833-f002:**
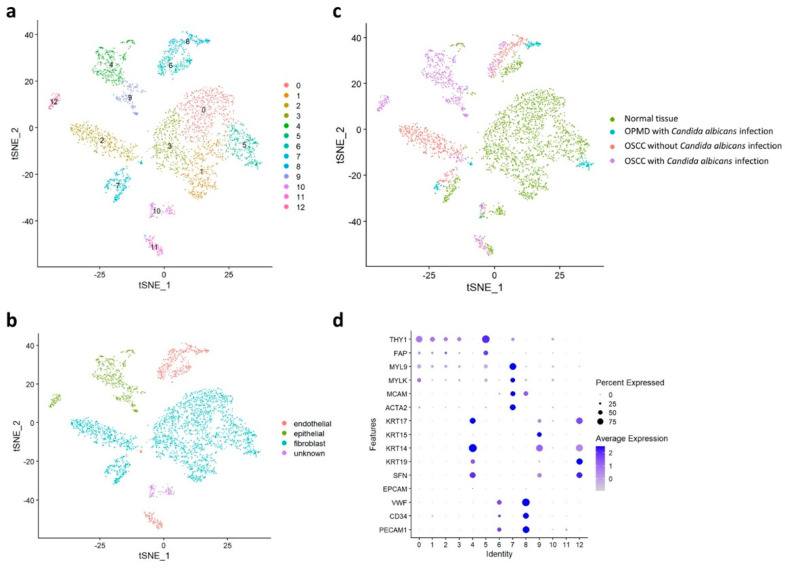
Cancer-related cell distribution in the normal tissue, OPMD lesion with *Candida albicans* infection, OSCC tissue with *C. albicans* infection, and OSCC tissue without *C. albicans* infection. (**a**) Classification of cancer−related cells into 13 subtypes by using the Louvain algorithm. (**b**) Classification of cancer-related cells into endothelial, epithelial, fibroblast, and unknown cells. (**c**) Separation of the four major groups. (**d**) Classification of cell types in each cluster by using specific biomarkers.

**Figure 3 ijms-23-04833-f003:**
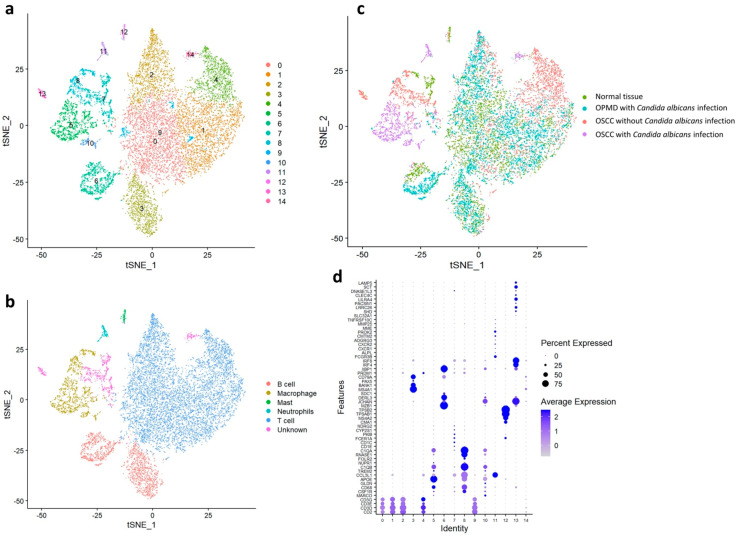
Immune cell distribution in the normal tissue, OPMD lesion with *Candida albicans* infection, OSCC tissue with *C. albicans* infection, and OSCC tissue without *C. albicans* infection. (**a**) Classification of immune cells into 15 subtypes by using the Louvain algorithm. (**b**) Classification of immune cells into B cells, macrophages, mast cells, neutrophils, T cells, and unknown cells. (**c**) Separation of the four major groups. (**d**) Classification of cell types in each cluster by using specific biomarkers.

**Figure 4 ijms-23-04833-f004:**
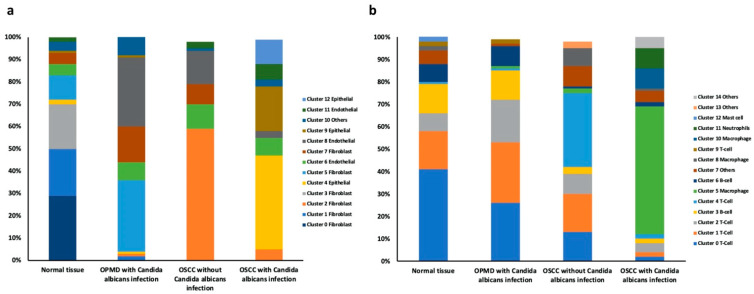
Comparison of the compositions of cell subtypes within the normal tissue, OPMD lesion with *Candida albicans* infection, OSCC tissue with *C. albicans* infection, and OSCC tissue without *C. albicans* infection. (**a**) Cancer-related cell ratios and (**b**) Immune cell ratios of these clusters within the four groups.

**Figure 5 ijms-23-04833-f005:**
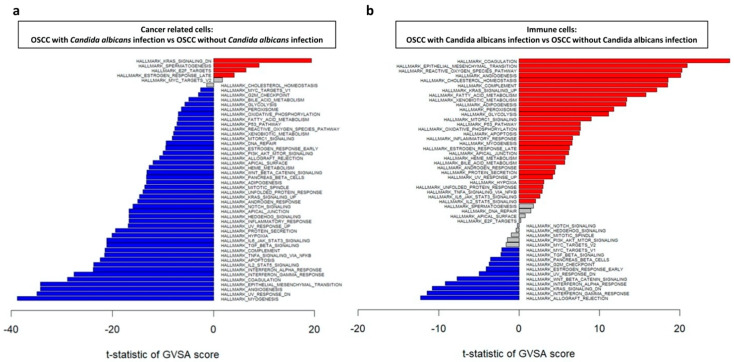
GSVA of regulatory pathways in (**a**) cancer-related cells and (**b**) immune cells in OSCC tissue with and without *Candida albicans* infection. Red and blue bars indicate that gene expression was significantly increased in OSCC tissue with *C. albicans* infection compared with that without *C. albicans* infection and OSCC tissue without *C. albicans* infection compared with that with *C. albicans* infection, respectively (both *p* < 0.05).

**Figure 6 ijms-23-04833-f006:**
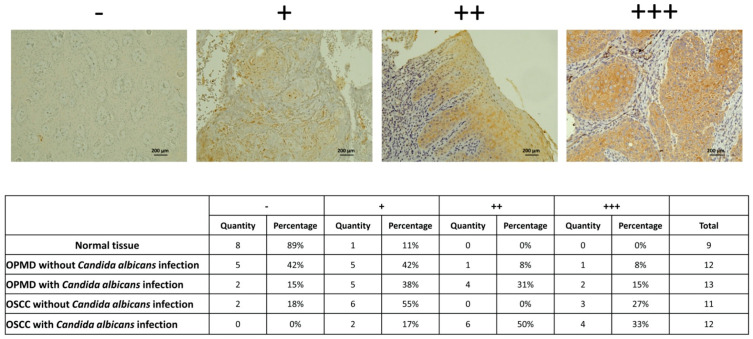
SFN significantly increased in OPMD lesions and OSCC tissues with *Candida albicans* infection. IHC staining data shows increased SFN expression in tissues with *C. albicans* infection, especially OSCC tissue. SFN staining intensity was interpreted by two researchers. −means negative stain; + means weak stain; ++ means moderate stain; +++ means strong stain.

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
