# Peer review of "Single-Cell RNA Sequencing Analysis for Oncogenic Mechanisms Underlying Oral Squamous Cell Carcinoma Carcinogenesis with *Candida albicans* Infection"

_ijms, 2022, doi:10.3390/ijms23094833_

Round 1
Reviewer 1 Report
The authors aimed to use, single-cell RNA sequencing (scRNA-seq) to investigate OSCC’s malignant transformation with and without C. albicans infection. Their results highlight the cell subpopulations that change during OSCC progression as well as the related cancer cell–TME interactions.
The study covers some issues that have been overlooked in other similar topics. The structure of the manuscript appears adequate and well divided in the sections. Moreover, the study is easy to follow, but few issues should be improved. Some of the comments that would improve the overall quality of the study are:
- Authors must pay attention to the technical terms acronyms they used in the text. Please better stated the aim of the study in the abstract section.
- English language needs to be revised.
- Limitations of the study needs to be added.
- Conclusion Section: This paragraph required a general revision to eliminate redundant sentences and to add some "take-home message".
Author Response
Thank you very much for your review of this manuscript.
Reviewer’s comments 1
Authors must pay attention to the technical terms acronyms they used in the text. Please better stated the aim of the study in the abstract section.
Response: The abstract has been revised the acronyms were annotated and the aim of the study was stated as below.
Abstract: Oral squamous cell carcinoma (OSCC) carcinogenesis involves heterogeneous tumor cells, and the tumor microenvironment (TME) is highly complex with many different cell types. Cancer cell–TME interactions are crucial in OSCC progression. Candida albicans (C. albicans )—frequently present in the oral potentially malignant disorder (OPMD) lesions and OSCC tissues—promotes malignant transformation. The aim of the study is to verify the mechanisms underlying OSCC carcinogenesis with C. albicans infection and identify the biomarker for the early detection of OSCC and as the treatment target. The single-cell RNA sequencing analysis (scRNA-seq) was performed to explore the cell subtypes in normal oral mucosa, OPMD, and OSCC tissues. The cell composition changes and oncogenic mechanisms underlying OSCC carcinogenesis with C. albicans infection were investigated. Gene Set Variation Analysis (GSVA) was used to survey the mechanisms underlying OSCC carcinogenesis with and without C. albicans infection. The results revealed specific cell clusters contributing to OSCC carcinogenesis with and without C. albicans infection. The major mechanisms involved in OSCC carcinogenesis without C. albicans infection are the IL2/STAT5, TNFα/NFκB, and TGFβ signaling pathways, whereas those involved in OSCC carcinogenesis with C. albicans infection are the KRAS signaling pathway and E2F target down-stream genes. Finally, stratifin (SFN) was validated to be a specific biomarker of OSCC with C. albicans infection. Thus, the detailed mechanism underlying OSCC carcinogenesis with C. albicans infection we determined and identified the treatment biomarker with potential precision medicine applications.
Reviewer’s comments 2
English language needs to be revised.
Response: The manuscript has been English edited by Wallace Academic Editing.
Reviewer’s comments 3
Limitations of the study needs to be added
Response: The limitation of the study has been revised in discussion as below.
In this research, there are only one specimen collected in each group of normal oral mucosa, OPMD with Candida albicans infection, OSCC with C. albicans infection, and OSCC without C. albicans infection. The individual genetic background could not be eliminated due to only one patient recruited in each group. However, due to omics data generated by scRNA-seq, a total of 18,433 cells were isolated for clustering, cell typing analyses, and identifying differentially expressed features. The major mechanisms involved in OSCC carcinogenesis with or without C. albicans infection could be identified. The other limitation of this research is our result could not be solidated due to C. albicans infection oral mucosa and OPMD lesion without C. albicans infection were not recruited. It is because no need of surgical intervention for the patient oral mucosa with C. albicans infection and difficult to find the OPMD lesion without C. albicans infection due to the prevalence of oral C. albicans infection is high in patients with OPMD [23]. However, the major mechanisms involved in OSCC carcinogenesis without C. albicans infection are the IL2/STAT5, TNFα/NFκB, and TGFβ signaling pathways, whereas those involved in OSCC carcinogenesis with C. albicans infection are the KRAS signaling pathway and E2F target downstream genes were revealed in this research. The stratifin (SFN) was also validated to be a specific biomarker of OSCC with C. albicans infection. The detailed mechanism underlying OSCC carcinogenesis with C. albicans infection was determined in this research and identified the treatment biomarker with potential precision medicine applications.
Reviewer’s comments 4
Conclusion Section: This paragraph required a general revision to eliminate redundant sentences and to add some "take-home message".
Response: The conclusion has been revised as below.
The specific cell clusters contribute to OSCC carcinogenesis with and without C. albicans infection were revealed in this research. The cluster 5 and 7 fibroblasts are involved in OPMD lesions with C. albicans infection, cluster 2 fibroblasts promote OSCC carcinogenesis without C. albicans infection, and cluster 4, 9, and 12 epithelial cells are major cell types in OSCC tissues with C. albicans infection. Furthermore, cluster 5 and 10 macrophages were found to play a crucial role in OSCC carcinogenesis with C. albicans infection. The GSVA data demonstrated that the IL2/STAT5, TNFα/NFκB, and TGFβ signaling pathways are the essential mechanisms promoting OSCC carcinogenesis without C. albicans infection and that the KRAS signaling pathway and E2F target downstream genes are the major mechanisms promoting OSCC carcinogenesis with C. albicans infection. The SFN was also validated to be a specific biomarker of OSCC with C. albicans infection.

Reviewer 2 Report
it is a well drafted manuscript. needs some minor revision
1) provide a full form for SFN
2) provide limitations and future perspective of the study
3) elaborate on from where normal tissue was taken
4) supplementary files are missing

Author Response
Thank you very much for your review of this manuscript.
Reviewer’s comments 1
English language and style are fine/minor spell check required
Response: The manuscript has been English edited by Wallace Academic Editing.
Reviewer’s comments 2
provide a full form for SFN
Response: Full form of stratifin (SFN)
stratifin [Homo sapiens]
GenBank: KAI2515796.1
Identical Proteins FASTA Graphics
LOCUS: KAI2515796, 248 aa, linear, PRI 12-APR-2022
DEFINITION: stratifin [Homo sapiens].
ACCESSION: KAI2515796
VERSION: KAI2515796.1
DBLINK: BioProject: PRJNA730525, BioSample: SAMN13957917
DBSOURCE: accession CM034951.1
ORIGIN
1 merasliqka klaeqaerye dmaafmkgav ekgeelscee rnllsvaykn vvggqraawr
61 vlssieqksn eegseekgpe vreyrekvet elqgvcdtvl glldshlike agdaesrvfy
121 lkmkgdyyry laevatgddk kriidsarsa yqeamdiskk empptnpirl glalnfsvfh
181 yeianspeea islakttfde amadlhtlse dsykdstlim qllrdnltlw tadnageegg
241 eapqepqs
Reviewer’s comments 3
provide limitations and future perspective of the study
Response: Limitations and future perspective of the study has been revised in discussion as below.
SFN is a member of the 14-3-3 family of highly conserved soluble acidic proteins that regulate various biological functions—which include cell cycling, growth, development, survival, and death as well as gene transcription. SFN, strongly expressed in many cancers, is involved in carcinogenesis through the regulation of cell proliferation, differentiation, and death in tumors [49, 50]. In the current study, SFN was found to be strongly expressed in the tissue with C. albicans infection, particularly the OSCC tissue. Studies have demonstrated that SFN is downstream of the E2F target gene and that SFN expression is relatively high in hepatocellular carcinoma cells with E2F overexpression and TP53 mutation [51, 52]. SFN may thus serve as the specific biomarker for predicting malignant transformation due to C. albicans infection. In the future, we will further explore the mechanism of SFN involves in the oral squamous cell carcinoma carcinogenesis with C. albicans infection. SFN has potential to develop as an OSCC biomarker, which could be developed for precision medicine applications.
In this research, there are only one specimen collected in each group of normal oral mucosa, OPMD with Candida albicans infection, OSCC with C. albicans infection, and OSCC without C. albicans infection. The individual genetic background could not be eliminated due to only one patient recruited in each group. However, due to omics data generated by scRNA-seq, a total of 18,433 cells were isolated for clustering, cell typing analyses, and identifying differentially expressed features. The major mechanisms involved in OSCC carcinogenesis with or without C. albicans infection could be identified. The other limitation of this research is our result could not be solidated due to C. albicans infection oral mucosa and OPMD lesion without C. albicans infection were not recruited. It is because no need of surgical intervention for the patient oral mucosa with C. albicans infection and difficult to find the OPMD lesion without C. albicans infection due to the prevalence of oral C. albicans infection is high in patients with OPMD [23]. However, the major mechanisms involved in OSCC carcinogenesis without C. albicans infection are the IL2/STAT5, TNFα/NFκB, and TGFβ signaling pathways, whereas those involved in OSCC carcinogenesis with C. albicans infection are the KRAS signaling pathway and E2F target downstream genes were revealed in this research. The stratifin (SFN) was also validated to be a specific biomarker of OSCC with C. albicans infection. The detailed mechanism underlying OSCC carcinogenesis with C. albicans infection was determined in this research and identified the treatment biomarker with potential precision medicine applications.
Reviewer’s comments 4
elaborate on from where normal tissue was taken
Response: Normal tissue was collected from the healthy patient who received wisdom teeth extraction.
Lesion locations and types of patient specimens were listed in the supplementary file.
Reviewer’s comments 5
supplementary files are missing
Response: We resubmit the supplementary file.

Reviewer 3 Report
In the present manuscript “Single-Cell RNA Sequencing Analysis for Oncogenic Mechanisms Underlying Oral Squamous Cell Carcinoma Carcinogenesis With Candida albicans Infection”, Yi-Ping Hsieh and collaborators make use of 10X-sc sequencing technology to analyze the molecular cell heterogeneity within OSCC lesions at different stages, and in particular focus on OSCC with and without Candida albicans infection. The use of this technology is interesting and relevant in the study of mechanism of tumoral progression because it can disaggregate the tumour heterogeneity that exists within the tumours and can confer specific properties to the specific cell types.
As an added value the authors have found a biological biomarker in OSCC infected with C. albicans, SFN, whose protein expression in cancer cells could potentially predict malignant transformation.
Even if the manuscript shows interesting potential, overall the computational analysis and interpretation is quite superficial for the claims of the authors. The manuscript is plenty of overstatements without scientifically rigor that would be advisable not to accept in the present form.
Major aspects to review
- The authors use a n=1 per group, what limit the conclusions that can be extracted from the work, especially when we are talking about samples coming from patients. This should be mentioned in the discussion of the manuscript.
Is the normal (non-transformed) tissue coming from a lesion (that is, normal adjacent tissue of a lesion) or is coming from a healthy oral mucosa? I had some doubts about this sample.
- In the computational analysis (Fig 1) the plot displays the Immune and the cancer-related cells clusters. Further they have done subclustering analysis (fig 1b,d). Can the authors clarify in the figure which is the identity of these cell subtypes? Or what they resemble? The specific biomarkers in Fig 1d, what they define and which is the source data of this classification? Also consider that resolution of the images is poor and the numbers or letter cannot be understood.
The tSNE plot in fig 1c doesn’t reflect that on Fig1a,b and one cannot check how the different samples distribute along cell types. (In fact this is the same figure that Figure 2c, in which only cancer-related cells are studied(?).
- Figure 2. Again, can the authors clarify in the figure which is the identity of these cell subtypes? Which is the molecular heterogeneity of the 3 different main cell types? Because the authors already have the data, it would be interesting for the publication at least annotate the different subclusters.
Why normal tissue display exclusively traits of fibroblasts? Epithelial cells are restricted to OSCC with C. albicans infection. Data seem to be a bit weird.
- Figure 3. It would be advisable try to improve the resolution of the figure because one cannot even read the named in Fig3d. I would suggest to annotate the main subclusters, specially the ones that seem to be relevant for the authors findings (macrophages, T cells…).
- Figures 4 and 5. The conclusions that the authors have made in the manuscript based on these results are overstated, for instance:
“…These results indicated that cluster 5 and 7 fibroblasts are involved in OPMD lesions with C. albicans infection, that cluster 2 fibroblasts promote OSCC carcinogenesis without C. albicans infection, and cluster 4, 9, and 12 epithelial cells are major cell types in OSCC tissues with C. albicans infection…”
“…These results revealed that cluster 4 T cells are involved in OSCC carcinogenesis without C. albicans infection. Moreover, the reduction in the number of cluster 0, 1, and 2 T cells was noted to be a factor contributing to the promotion of malignant transformation due to C. albicans infection. Furthermore, cluster 5 and 10 macrophages were found to play a crucial role in OSCC carcinogenesis with C.albicans infection…”
The authors should note that they have not develop in this work any functional assay to corroborate what they are claiming and these assertions need to be cut down in the context of the presented data.
Also consider that if one subcell type is more/less abundant in one sample that in other, doesn’t necessary mean that is involved in something relevant for the carcinogenic process.
“…Taken together, these results revealed that the mechanisms underlying OSCC carcinogenesis due to C. albicans infection are different from those underlying OSCC carcinogenesis without C. albicans infection. The IL2/STAT5, TNFα/NFκB, and TGFβ signaling pathways are essential for promoting OSCC carcinogenesis without C. albicans infection; moreover, immune cells with high expression of KRAS, WNT/β-catenin, and TGFβ signaling pathway genes contribute to this promotion…”
The same reasons, the authors should note that they have not develop in this work any functional assay to corroborate what they are claiming.
It would be more interesting for publication to do this type of analysis within the specific subclusters, check which are the subclusters with more transcriptional changes and indagate within them.
It would be also interesting to do the GSVA study comparing normal tissue vs OPMD and OPMD vs OSCC.
- Figure 6. These data have potential clinical interest. I would suggest to clarify a bit better the figure and the figure legend. Maybe adding the labels to the pics and explaining in the figure legend what the numbers mean.
Author Response
Thank you very much for your review of this manuscript.
Reviewer’s comments 1
The authors use a n=1 per group, what limit the conclusions that can be extracted from the work, especially when we are talking about samples coming from patients. This should be mentioned in the discussion of the manuscript.
Response: The limitation of the study has been revised in discussion as below.
In this research, there are only one specimen collected in each group of normal oral mucosa, OPMD with Candida albicans infection, OSCC with C. albicans infection, and OSCC without C. albicans infection. The individual genetic background could not be eliminated due to only one patient recruited in each group. However, due to omics data generated by scRNA-seq, a total of 18,433 cells were isolated for clustering, cell typing analyses, and identifying differentially expressed features. The major mechanisms involved in OSCC carcinogenesis with or without C. albicans infection could be identified. The other limitation of this research is our result could not be solidated due to C. albicans infection oral mucosa and OPMD lesion without C. albicans infection were not recruited. It is because no need of surgical intervention for the patient oral mucosa with C. albicans infection and difficult to find the OPMD lesion without C. albicans infection due to the prevalence of oral C. albicans infection is high in patients with OPMD [23]. However, the major mechanisms involved in OSCC carcinogenesis without C. albicans infection are the IL2/STAT5, TNFα/NFκB, and TGFβ signaling pathways, whereas those involved in OSCC carcinogenesis with C. albicans infection are the KRAS signaling pathway and E2F target downstream genes were revealed in this research. The stratifin (SFN) was also validated to be a specific biomarker of OSCC with C. albicans infection. The detailed mechanism underlying OSCC carcinogenesis with C. albicans infection was determined in this research and identified the treatment biomarker with potential precision medicine applications.
Reviewer’s comments 2
Is the normal (non-transformed) tissue coming from a lesion (that is, normal adjacent tissue of a lesion) or is coming from a healthy oral mucosa? I had some doubts about this sample.
Response: Normal tissue is the healthy gingival mucosa was collected when the healthy patient received wisdom teeth extraction.
We have revised the types of patient specimens in the material and method as below.
4.1. Clinical Patient Specimen Collection
This study was approved by the Institutional Review Board of National Cheng Kung University Hospital (IRB No.: B-ER-108-424) and complied with medical research protocols outlined in the Declaration of Helsinki by the World Medical Association.
The included patients underwent scheduled excision surgery. We collected one specimen each of the OPMD lesion with C. albicans infection, and OSCC tissue with or without C. albicans infection. Normal tissue is the healthy gingival mucosa which was collected when the healthy patient received wisdom teeth extraction. Thereafter, tissue dissociation and further analysis were performed.
Lesion locations and types of patient specimens were listed in the supplementary file.
Reviewer’s comments 3
In the computational analysis (Fig 1) the plot displays the Immune and the cancer-related cells clusters. Further they have done subclustering analysis (fig 1b,d). Can the authors clarify in the figure which is the identity of these cell subtypes? Or what they resemble? The specific biomarkers in Fig 1d, what they define and which is the source data of this classification? Also consider that resolution of the images is poor and the numbers or letter cannot be understood.
Response: According to the t-distributed stochastic neighbor embedding (tSNE) results, these cells could be divided into immune cells and cancer-related cells according to their typical biomarkers. Clusters 2, 7, 12, 13, 16, and 19 included cancer-related cells, whereas the remaining 17 clusters contained immune cells. The cell types of these clusters were divided in the Fig 2. and Fig 3. in detail.
The specific biomarkers used for the division of cell types refer to the reference as below.
- V. Puram, I. Tirosh, A. S. Parikh, A. P. Patel, K. Yizhak, S. Gillespie, C. Rodman, C. L. Luo, E. A. Mroz, K. S. Emerick, D. G. Deschler, M. A. Varvares, R. Mylvaganam, O. Rozenblatt-Rosen, J. W. Rocco, W. C. Faquin, D. T. Lin, A. Regev and B. E. Bernstein. Single-Cell Transcriptomic Analysis of Primary and Metastatic Tumor Ecosystems in Head and Neck Cancer. Cell 2017, 171, 1611-1624.e24. 10.1016/j.cell.2017.10.044
We have revised the figure and replaced with the higher resolution of the images in the manuscript.
Reviewer’s comments 4
The tSNE plot in fig 1c doesn’t reflect that on Fig1a,b and one cannot check how the different samples distribute along cell types. (In fact this is the same figure that Figure 2c, in which only cancer-related cells are studied(?).
Response: Fig 1c have been revised. Cancer-related cell distribution in the normal tissue, OPMD lesion with Candida albicans infection, OSCC tissue with C. albicans infection, and OSCC tissue without C. albicans infection were shown as Figure 2. Immune cell distribution were shown as Figure 3.
Reviewer’s comments 5
Figure 2. Again, can the authors clarify in the figure which is the identity of these cell subtypes? Which is the molecular heterogeneity of the 3 different main cell types? Because the authors already have the data, it would be interesting for the publication at least annotate the different subclusters.
Why normal tissue display exclusively traits of fibroblasts? Epithelial cells are restricted to OSCC with C. albicans infection. Data seem to be a bit weird
Response: Cells were divided into different clusters according to gene expression profiles in the scRNA-seq results. The same cell types may express different gene expression profiles; therefore, they may be divided into different clusters.
The compositions of cell subtypes in the normal tissue were listed in the supplementary Table 2. Normal tissue, and OSCC with C. albicans infection composed of fibroblast, epithelial and endothelial cells with different percentage.
The compositions of cell subtypes in the group were divided according to specific biomarkers of cell types.
Reviewer’s comments 6
Figure 3. It would be advisable try to improve the resolution of the figure because one cannot even read the named in Fig3d. I would suggest to annotate the main subclusters, specially the ones that seem to be relevant for the authors findings (macrophages, T cells…).
Response: We have revised the figure and replaced with the higher resolution of the images in the manuscript.
Reviewer’s comments 7
Figures 4 and 5. The conclusions that the authors have made in the manuscript based on these results are overstated, for instance:
“…These results indicated that cluster 5 and 7 fibroblasts are involved in OPMD lesions with C. albicans infection, that cluster 2 fibroblasts promote OSCC carcinogenesis without C. albicans infection, and cluster 4, 9, and 12 epithelial cells are major cell types in OSCC tissues with C. albicans infection…”
Response: According to the result of the compositions of cell subtypes in the groups, the cluster 5 and 7 fibroblasts were significantly increased in the OPMD lesions with C. albicans infection compared to normal tissue. The cluster 5 and 7 fibroblasts may involve in OPMD development.
Cluster 2 fibroblasts significantly increased in the OSCC without C. albicans infection compared to normal tissue. The cluster 2 may play an important role in OSCC carcinogenesis without C. albicans infection.
We have revised the description of our result as below.
These results indicated that cluster 5 and 7 fibroblasts were related to the malignant transformation in OPMD lesions with C. albicans infection, that cluster 2 fibroblasts may play an important role in OSCC carcinogenesis without C. albicans infection, and cluster 4, 9, and 12 epithelial cells are major cell types in OSCC tissues with C. albicans infection
Reviewer’s comments 8
“…These results revealed that cluster 4 T cells are involved in OSCC carcinogenesis without C. albicans infection. Moreover, the reduction in the number of cluster 0, 1, and 2 T cells was noted to be a factor contributing to the promotion of malignant transformation due to C. albicans infection. Furthermore, cluster 5 and 10 macrophages were found to play a crucial role in OSCC carcinogenesis with C.albicans infection…”
Response: According to the result of the compositions of cell subtypes in the groups, the cluster 4 T cells significantly increased in the OSCC with C. albicans infection compared to normal tissue. The cluster 4 T cells may play an important role in OSCC carcinogenesis without C. albicans infection. Cluster 0, 1, and 2 T cells significantly decreased in the OSCC with C. albicans infection compared to normal tissue. The reduction in the number of cluster 0, 1, and 2 T cells may contribute to the promotion of malignant transformation. Cluster 5 and 10 macrophages significantly increased in the OSCC with C. albicans infection compared to normal tissue. The results showed cluster 5 and 10 macrophages may play a crucial role in OSCC carcinogenesis with C. albicans infection.
We have revised the description of our result as below.
These results revealed that cluster 4 T cells may correlate with OSCC carcinogenesis without C. albicans infection. Moreover, the reduction in the number of cluster 0, 1, and 2 T cells may be a factor contributing to the promotion of malignant transformation due to C. albicans infection. Furthermore, cluster 5 and 10 macrophages may play a crucial role in OSCC carcinogenesis with C. albicans infection.
Reviewer’s comments 9
“…Taken together, these results revealed that the mechanisms underlying OSCC carcinogenesis due to C. albicans infection are different from those underlying OSCC carcinogenesis without C. albicans infection. The IL2/STAT5, TNFα/NFκB, and TGFβ signaling pathways are essential for promoting OSCC carcinogenesis without C. albicans infection; moreover, immune cells with high expression of KRAS, WNT/β-catenin, and TGFβ signaling pathway genes contribute to this promotion…”
Response: According to the result of GSVA in the groups, the IL2/STAT5, TNFα/NFκB, and TGFβ signaling pathways were increased in the OSCC without C. albicans infection. In immune cells, the expression of KRAS, WNT/β-catenin, and TGFβ signaling pathway genes were increased in the OSCC without C. albicans infection. These pathways may play important role in regulating OSCC carcinogenesis.
We have revised the description of our result as below.
Taken together, these results revealed that the mechanisms underlying OSCC carcinogenesis due to C. albicans infection are different from those underlying OSCC carcinogenesis without C. albicans infection. The IL2/STAT5, TNFα/NFκB, and TGFβ signaling pathways may be essential for promoting OSCC carcinogenesis without C. albicans infection; moreover, immune cells with high expression of KRAS, WNT/β-catenin, and TGFβ signaling pathway genes may contribute to this promotion. On the contrary, in OSCC carcinogenesis with C. albicans infection, the KRAS signaling pathway and E2F target downstream genes may be the major mechanisms involved; here, the upregulation of ROS and KRAS signaling pathway genes in immune cells corelated to the promotion of OSCC progression.
Reviewer’s comments 10
It would be more interesting for publication to do this type of analysis within the specific subclusters, check which are the subclusters with more transcriptional changes and indagate within them.
Response: Thank you very much for your suggestion. We will perform the analysis within the specific subclusters and more transcriptional changes in detail in the future.
Reviewer’s comments 11
It would be also interesting to do the GSVA study comparing normal tissue vs OPMD and OPMD vs OSCC.
Response: We performed the GSVA comparing candida OSCC vs candida OPMD, candida OSCC vs non-candida OSCC, and non-candida OSCC vs normal tissue. The data were revised in the supplementary data.
Reviewer’s comments 12
Figure 6. These data have potential clinical interest. I would suggest clarifying a bit better the figure and the figure legend. Maybe adding the labels to the pics and explaining in the figure legend what the numbers mean.
Response: We have revised the manuscript and adding the labels to the pics and explaining in the figure legend what the numbers mean.

Round 2
Reviewer 3 Report
Even if the authors have made some changes in the text within the manuscript there is no solid basis for the claims of the paper. The main concerns still there are.